# Implementing a Multi-Disciplinary, Evidence-Based Resilience Intervention for Moral Injury Syndrome: Systemic Barriers and Facilitators

**DOI:** 10.3390/bs14040281

**Published:** 2024-03-28

**Authors:** J. Irene Harris, Shawn Dunlap, Danielle Xanthos, Jeffrey M. Pyne, Eric Hermes, Brandon J. Griffin, Susannah Robb Kondrath, Se Yun Kim, Kristin B. Golden, Nathaniel J. Cooney, Timothy J. Usset

**Affiliations:** 1VA Maine Healthcare System, Augusta, ME 04330, USA; susannah.kondrath@va.gov (S.R.K.); nathaniel.cooney@va.gov (N.J.C.); timothy.usset@va.gov (T.J.U.); 2Department of Psychology, University of Maine, Orono, ME 04469, USA; 3Center for Healthcare Organization and Implementation Research, VA Bedford Healthcare System, Bedford, MA 01730, USA; shawn.dunlap@va.gov (S.D.); kristin.godlin@gmail.com (K.B.G.); 4Northport VA Medical Center, Northport, NY 11768, USA; danielle.xanthos@va.gov; 5Center for Mental Health Outcomes Research, Central Arkansas Veterans Healthcare System, North Little Rock, AR 72114, USA; jeffrey.pyne@va.gov (J.M.P.);; 6Psychiatric Research Institute, University of Arkansas for Medical Sciences, Little Rock, AR 72205, USA; 7Veterans Affairs Northeast Program Evaluation Center, Orange, CT 06516, USA; 8Division of Health Policy and Management, University of Minnesota, Minneapolis, MN 55455, USA

**Keywords:** moral injury syndrome, moral injury, veterans, spirituality, spiritually integrated care, Consolidated Framework for Implementation Research, implementation, dissemination, qualitative research

## Abstract

Moral injury syndrome (MIS) is a mental health (MH) problem that substantially affects resilience; the presence of MIS reduces responsiveness to psychotherapy and increases suicide risk. Evidence-based treatment for MIS is available; however, it often goes untreated. This project uses principles of the Consolidated Framework for Implementation Research (CFIR) to assess barriers and facilitators to the implementation of Building Spiritual Strength (BSS), a multi-disciplinary treatment for MIS. Interviews were conducted with chaplains and mental health providers who had completed BSS facilitator training at six sites in the VA. Data were analyzed using the Hamilton Rapid Turnaround method. Findings included multiple facilitators to the implementation of BSS, including its accessibility and appeal to VA chaplains; leadership by VA chaplains trained in the intervention; and effective collaboration between the chaplains and mental health providers. Barriers to the implementation of BSS included challenges in engaging mental health providers and incorporating them as group leaders, veterans’ lack of familiarity with the group format of BSS, and the impact of the COVID-19 pandemic. Results highlight the need for increased trust and collaboration between VA chaplains and mental health providers in the implementation of BSS and treatment of MIS.

## 1. Introduction

In the United States (US) military, service members may experience events that are morally injurious. They may do things that violate their personal moral or ethical standards. They may also have witnessed or been the victim of others violating these standards [1]. If distress about these situations is unresolved, it can elicit moral injury syndrome (MIS) [2,3]. Symptoms of MIS include guilt, hopelessness, sadness, self-blame, powerlessness, social withdrawal, suicidal ideation and suicide attempts, re-experiencing, anger, and spiritual distress [2,3,4,5,6,7,8].

MIS is often co-morbid with posttraumatic stress disorder (PTSD) in veterans; 35–60% of those seeking treatment for PTSD have comorbid MIS [1,9]. Individuals who experience both MIS and another mental health (MH) problem, like PTSD, generally have more severe symptoms, a longer course of disorder, and benefit less from treatment [7,9,10]. MIS can also exist independently of PTSD [6,11]. PTSD is associated with exposure to events that threaten life or physical integrity, and symptoms include biological and behavioral manifestations of fear. Moral injury is associated with exposure to events that challenge core moral beliefs and is expressed in values/spirituality/meaning systems, intractable guilt/anger, hopelessness, social withdrawal, and ruminative attempts to make meaning [3].

Critically, the presence of MIS greatly increases suicide risk [5,6,12]. One study of a nationally representative sample of US post-9/11 veterans found that male veterans who endorsed transgressing their own values were up to 100% more likely to attempt suicide, even after controlling for sociodemographic characteristics, pre-military history of suicidal ideation and suicide attempts, and current mental health status (e.g., PTSD, depression, alcohol misuse) [12]. Veteran women exposed to a potentially morally injurious event (PMIE) were also 50% more likely to report a suicide attempt during and after military service, particularly if they endorsed being betrayed by their military leaders or peers [12]. Thus, it is important that suicide prevention programming consider the impact of moral injury on veterans, especially as they transition from military to civilian life.

Despite the estimated prevalence of MIS among US military veterans and its significant impact on clinical outcomes, evidence suggests that most VA facilities do not provide evidence based treatment for MIS [13]. There are several reasons why MIS may go untreated within VA:MIS is not included in the DSM-V-TR or the ICD10. There are no billing codes for MIS, and no consensus exists on psychometric measurement.The VA has not yet identified any specific treatment(s) as a gold standard treatment for MIS.Many healthcare providers receive little training in MIS and related spiritual concerns and are uncomfortable with the spiritual language many veterans use to describe MIS [14,15].Many veterans feel they cannot talk about moral injury symptoms in psychotherapy for PTSD [16].

Given that our ability to measure moral injury syndrome (not just exposure to potentially morally injurious events) is new, the current literature is not conclusive about the effectiveness of first-line PTSD treatments when moral injury syndrome is co-morbid [16,17,18]. A number of treatments designed to address MIS have been developed and tested in randomized controlled trials (RCTs) as treatments designed specifically for component symptoms of MIS. These include Adaptive Disclosure (a treatment for PTSD with a moral injury module [19,20]), Impact of Killing in War [21], Trauma-Informed Guilt Reduction [22], Building Spiritual Strength [23,24], The Self-Forgiveness Workbook [25], and Moral Elevation [26]. Because until very recently, measures of MIS symptoms did not exist, most studies of treatments for moral injury have used measures of PTSD as primary outcomes; this may not fully reflect the impact of these treatments on MIS [6].

Given that empirically supported and effective treatment for component symptoms of MIS is available, increasing adoption of such treatments for MIS is critical to protect veterans from the serious consequences of untreated MIS. The goal of this implementation project is to use Consolidated Framework for Implementation Research (CFIR) principles to assess barriers and facilitators to the implementation of Building Spiritual Strength (BSS), an evidence-based intervention for moral injury [27,28]. To date, 20 VA healthcare systems are providing BSS services for MIS, and another 60 have sought training to implement the program. By studying the BSS adoption process, it is possible to discover key facilitative and hindering factors in the adoption of an evidence-based treatment specific to moral injury. This will inform the process of making all empirically supported models of moral injury treatment available to veterans.

## 2. Method

### 2.1. Theoretical Foundation

The CFIR integrates previously published implementation theories to provide a more thorough and standardized understanding of interacting complex contextual factors by suggesting consistent terms and definitions applicable across diverse settings [28,29]. The CFIR is used to guide stages of data collection, analysis, and interpretation, allowing for greater effectiveness of implementation strategies, increased generalizability across a variety of contextual settings, and greater standardization when building on findings of prior studies. The CFIR identifies several major domains of influence: intervention characteristics, individual characteristics, organizational factors, and process [27]. The framework also determines the strength of the impact of these constructs in high and low implementation settings. The CFIR promotes the development of effective implementation strategies based on the mitigation of identified barriers and the promotion of identified facilitators [29].

### 2.2. Design

This project used key informant interviews from seven VA clinicians who had completed BSS facilitator training. Interviews took place between June 2020 and November 2022. The clinicians represented both chaplains and mental health professionals at six sites. Sites were selected to represent (a) VA facilities and Veteran Centers that had very successful BSS programs (N = 3 sites) and (b) VA facilities and Veteran Centers that failed to implement a successful BSS program (N = 3 sites). One site provided two interviewees, and the rest provided one interviewee each. Participants were interviewed by a BSS facilitator trainer who had provided the training at each site. Thus, the interviewer was aware of each participant’s role in the organizational context. A research assistant also attended each session to transcribe responses. The qualitative interview used was organized based on CFIR principles (see Appendix A). The facility institutional research board reviewed the project and determined that it was not research, approving it as exempt.

### 2.3. Sites

Sites were selected to represent a broad range of regional cultures, facility sizes, and provider disciplines. In the course of routine follow-up contact to individuals who had attended BSS facilitator training, the BSS leadership team invited facilitators from sites that were either clearly very successful, or clearly unsuccessful, in establishing a BSS program. “Very successful” sites were those that were able to provide multiple cohorts of veterans with BSS services. “Clearly unsuccessful sites” were those in which either no BSS groups were offered or the program was ended after only one cohort. Participation in interviews for this project were voluntary and confidential. ***Site 1*** was an urban, Southeastern VA medical center. At that site, a chaplain and social worker co-led BSS groups. The site had recently started providing BSS services, and eight veterans had completed the program. Measurement-based care data indicated that veterans in the group had reduced symptoms of depression and spiritual distress and improved quality of life. ***Site 2*** was an urban, Northwestern VA medical center. At that site, a single chaplain had been leading BSS groups for four years, and 208 veterans had completed the group. Measurement-based care data were not available for this site. ***Site 3*** was a smaller, urban VA medical center in the northern Midwest that located BSS groups in local Vet Centers. This site had been using BSS for five years, and 48 veterans had completed BSS. Chaplain trainees provided BSS services at this site, with chaplain and social work supervisors who were also trained as BSS facilitators. Measurement-based care data were not available for this site. ***Site 4*** was a rural, Northeastern VA medical center. A chaplain and a social worker had trained to be BSS facilitators, but the social worker was not able to add the program to their workload. The site had not initiated BSS services. ***Site 5*** was a rural, Southern VA medical center, in which chaplain trainees had BSS training; no other chaplaincy or mental health staff had been trained. The site provided one BSS group cohort and then ended the program. ***Site 6*** was a very rural, northern Midwest VA medical center. Several mental health providers and chaplains attended BSS training, but the facility had engaged in BSS training with the goal of generally informing staff about MIS, rather than a goal of initiating BSS services. 

### 2.4. Participants

The seven participants included two women and five men. Four identified as White, one as Asian American, one as African American, and one as Latinx. At least two participants identified as religious minorities. The sample included four chaplains, two social workers, and one psychologist. With the exception of two chaplain administrators (chiefs of chaplains), all were full-time direct care providers in their discipline.

When comparing participant sites to one another, as well as data from 10 sites that provided program outcome information to the BSS leadership team from 2015 to 2020, it appears that chaplains, and chaplains collaborating with mental health providers, were most successful in creating BSS programs. To our knowledge, there is only one site in the nation in which a mental health provider has started a successful BSS program without chaplaincy collaboration. Also, while collaboration between chaplaincy and mental health appears to facilitate implementation, the program that brought BSS to the most veterans was administered entirely by a single chaplain. When the race of program facilitators and administrators was considered, facilitators and administrators who were Black, Indigenous, or People of Color (BIPOC) were quite successful in establishing programs and steadily recruiting veterans into groups, as well as collecting outcome data. When the professional training of BSS facilitators and administrators was considered, it was noted that two sites in the nation only had chaplain residents or psychology interns receive training in BSS. (One of these was Site 5.) Neither of these sites were successful in sustained implementation of BSS services. When the location of sites was considered, urban VA medical centers appear to have been more successful than rural VA medical centers in implementing BSS.

### 2.5. Measures (Interview Guide)

The interview guide was designed to reflect the CFIR identified domains, intervention characteristics, individual characteristics, organizational factors, and the implementation process. *Intervention characteristics* inquiries included reasons for facility or personal interest in BSS, what kind of evidence is needed to support implementing BSS, how BSS compares to other interventions used in the facility, what types of changes were made to BSS in implementation, and how complex BSS implementation was. *Individual characteristics* inquiries included disciplines and roles for trained BSS facilitators and presence and volume of BSS services provided at each site. *Organizational factors* inquiries included the structure and culture of the facility, who was involved in implementation, receptivity to BSS, and perceived need for BSS. *Implementation process* inquiries included whether BSS was being used as planned, perceptions of the BSS facilitator training process, incentives to implement BSS, access to needed resources for BSS, and factors outside the organization that impacted implementation.

### 2.6. Analysis

Data were analyzed using the Hamilton Rapid Turnaround method [30,31]. As recommended for rapid turnaround analytic approaches [32], the material was initially organized into a grid based on CFIR components by the interviewers, who represented two different disciplines (psychology and anthropology), then reviewed by the full research team (including psychologists, anthropologists, a physician, and a cross-trained psychologist/clergyperson) to develop consensus on results (see Table 1). After the psychologist and anthropologist team created the initial grid, the results were further reviewed by the remainder of the team. Differences in interpretation were resolved by consensus. The resulting grid was organized by intervention characteristics (why BSS?, evidence needed, changes needed, effectiveness), individual characteristics (disciplines trained, group leaders, confidence), organizational factors (team involved, organizational receptivity, leadership receptivity, organizational openness to innovation, perceived need for BSS, fit with organizational process), and implementation process (using as planned?, training, incentive, resource needs, factors outside of organization).

## 3. Results

Results compare and contrast responses from successful vs. unsuccessful sites, in order to identify factors that facilitate effective BSS implementation, as well as factors that act as barriers to BSS implementation. Facilitators and barriers are presented by the CFIR domain that emerged in several areas, including (a) personnel involved in advocating for, implementing, and sustaining BSS; (b) local effectiveness data; (c) training; (d) institutional expectations for cohort vs. open groups; (e) perceived need for interventions specific to MIS; and (f) the impact of the pandemic on implementation.

### 3.1. Intervention

#### 3.1.1. Why BSS?

The respondents at the sites that chose to implement BSS told us that they did so because of the BSS intervention’s unique focus, among group protocols for chaplains, on moral injury and PTSD, specifically noting its “more welcoming, more universal language.”

#### 3.1.2. Evidence Needed

While BSS is an evidence-based practice, the evidence supporting BSS was challenged by mental health providers at several facilities. When challenged about evidence supporting BSS, many chaplains reported that it was a struggle for them to describe the background research in terms meaningful to mental health providers. Some described a perception among mental health providers that any initiative involving chaplaincy could not be evidence-based. The barrier was confounded when the SARS-CoV-2 (COVID-19) pandemic necessitated remote or virtual groups. Subsequently, local BSS leaders were not able to collect outcome measures of PTSD, depression, or symptoms of moral injury for measurement-based care practices; pre-pandemic measurement-based care was a recommended practice for BSS programs. Lack of local effectiveness data made it even more difficult to demonstrate effectiveness to skeptical mental health providers. One approach which overcame this barrier at some sites was finding and including a BSS-trained advocate to champion the intervention in pitches aimed at mental health leadership. Participants told us that a local mental health champion was integral to making a “strong pitch” to decision-makers, allowing them “to buy-in on collaborations” between mental health services and chaplain services.

#### 3.1.3. Changes Needed

Several sites chose to supplement BSS in ways that enhanced adherence from veteran participants. These strategies included adding mindfulness exercises at the beginning of each session, adding additional sessions on the topic of forgiveness, as well as adding follow-up sessions after the standard intervention was completed. Interviewees also expressed that the follow-ups helped to “maintain therapeutic effects over a longer term.”

#### 3.1.4. Local Effectiveness

Another characteristic of BSS that affected implementation was its perceived effectiveness; successful sites viewed BSS as a useful suicide prevention tool, and they developed local psychometric and anecdotal program evaluation data. Veterans across sites reported BSS to be “healing” or “helpful”. For example, Site 1 maintained a database including both pre- and post-intervention scores on measures of depression, spiritual distress, and moral distress, as well as quotes from veterans who participated in BSS. These sites were able to use such information to support continued resources to the BSS program.

### 3.2. Individuals

#### 3.2.1. Disciplines Trained

Successful BSS implementation was associated with the backgrounds of the BSS facilitator trainees at each site. At most sites, both chaplains and mental health providers were trained to administer and co-facilitate BSS. At most successful sites, mental health providers bought into and understood the effectiveness and research background for BSS.

There were three instances of chaplains receiving BSS facilitator training at one facility and then subsequently developing BSS programs when they moved to another facility; these chaplains were typically effective in engaging mental health providers to participate in BSS facilitator training. While collaboration across mental health and chaplaincy seemed to be important in initiating and maintaining BSS programs, in most cases the initiation of developing a BSS program came from chaplaincy. One participant [supervisory chaplain] went as far as to say, “all chaplains should be trained in BSS”.

#### 3.2.2. Leaders

Successful sites were more likely to use the chaplain and mental health provider co-facilitator strategy. The available data point to co-facilitation as the ideal model for BSS groups. That said, there have been several examples of individual chaplains or mental health providers successfully leading BSS programs solo. For example, at Site #2, BSS was led only by a chaplain, and Site 2 provided BSS services to more veterans than any other site (over 200 veterans). At Site #2, BSS was adapted for use in residential programs for individuals managing PTSD with comorbid substance use disorder. Another site in the Northwest region, not sampled in this project, similarly provided chaplain-only BSS services for 185 veterans in residential treatment for PTSD and comorbid substance use disorder. More research is needed on the characteristics of those individuals and their facilities to identify the more specific facilitators for their success.

#### 3.2.3. Confidence

There were individual characteristics that acted as barriers to BSS implementation. Many of these had to do with BSS facilitator trainees; while some sites had chaplain and mental health trainees and licensed/certified providers working together to lead BSS groups, some sites used BSS training to meet didactic requirements for chaplaincy, psychology, or social work trainees. None of the sites that had only residents/interns trained subsequently implemented BSS programs; one participant characterized chaplaincy students as “apprehensive” about leading BSS groups without help. Another trainee-related barrier involved chaplaincy trainees who were struggling with inclusive theological approaches necessary for chaplaincy practice in a public medical setting, such as a VA medical center. One site reported that a student chaplain was either unable or unwilling to adhere to the manual after training because the student’s personal religious beliefs were inconsistent with inclusive chaplaincy practice.

Self-consciousness often reflected the practitioner’s relationship with the content of BSS. As one social worker told us, “I was not confident; I had not worked with spiritually integrated care before. I was self-conscious about spiritually integrated care because I am not Christian, not because the training was insufficient.” A chaplain specializing in mental health care delivery reported, “I was very confident about using BSS.” This chaplain had experience at multiple public health care sites and had specialized training in mental health chaplaincy. Even so, all of the sites that were unsuccessful in establishing a BSS program expressed a desire for more ongoing consultation/supervision in the use of the intervention.

### 3.3. Organization

#### 3.3.1. Team Involved

The most successful BSS leadership teams were multi-disciplinary, incorporating mental health and chaplaincy, often with the assistance of Local Recovery Coordinators and Suicide Prevention Coordinators. Local Recovery Coordinators were engaged through a national presentation to the Local Recovery Coordinator community, describing BSS as an option for veteran-centered, recovery-oriented, strengths-focused care. Suicide Prevention Coordinators became engaged through close collaboration with chaplaincy in a Community Clergy Training Program [33], which provided community clergy with information on suicide prevention, mental health, and spiritual care resources in the VA and in their communities.

#### 3.3.2. Organizational Receptivity

One site that met barriers to implementing BSS at a VA medical center was successful in establishing the program at a Vet Center. Chaplains perceived and experienced barriers within facilities that they suggested “did not value chaplaincy”, with some chaplains suggesting that the buy-in from both facility leadership and mental health (most frequently psychology) was only superficial.

Chaplain participants suggested further education on MIS and BSS for MH providers at reluctant sites. At one site, a veteran complained that the student chaplain who led the group was not using inclusive practices; this led to the end of the group without any consultation involving the student chaplains organizing it. In this case, the chaplain also perceived the issue to be at a higher level, saying “psychologists were threatened by [chaplains providing] moral injury treatment…trying to horn in on what they were doing.” This points to the need for collaborative conversations and role definitions between mental health and chaplaincy at sites seeking to implement BSS or other forms of spiritually integrated care. Both mental health providers and chaplains have unique skill sets relevant to care for MIS.

#### 3.3.3. Leadership Receptivity

Attending BSS training often helped group leaders make the case for BSS to mental health leadership. At some sites, chaplaincy supervisors were already aware of BSS through the National Chaplain Service and were thus very supportive of BSS implementation. At one site, a chaplain leader who was interested in BSS implementation left the site, making it more difficult to finish implementation processes. At least one psychologist recommended that when facilities participate in BSS trainings, leaders in Mental Health and Chaplaincy should be held responsible for seeing that the training is then used to initiate a BSS program.

#### 3.3.4. Organizational Openness to Innovation

Many mental health providers expressed gratitude for training in techniques for working with spirituality both ethically and appropriately. However, some chaplains reported receiving mixed messages about the value of being trained in BSS. For example, chaplains reported that they were encouraged to seek BSS training by the National Chaplain Service and by other chaplains but noted that chaplains who had developed other models for moral injury care sometimes discouraged them from pursuing BSS training.

#### 3.3.5. Perceived Need for BSS

Perceived need for interventions like BSS was similar across both successful and unsuccessful sites, with sites comparing it to PTSD regimens with more well-established footprints like Prolonged Exposure (PE), Cognitive Processing Therapy (CPT), and Cognitive Behavioral Therapy (CBT). BSS was often viewed as preferable to those options for patients who rejected or benefitted only minimally from established PTSD treatments. Several participants foresaw an expansion of MIS treatments to healthcare workers, as the effects of COVID-19 on this population is documented in the literature. The group format was cited as “necessary for healing” because the expected mechanism of action in BSS is increasing the capacity to recognize moral complexity; thus, having members of a group share different perspectives was seen as necessary to help veterans integrate multiple perspectives related to the same moral problem. Some sites reported that regional cultural differences helped to facilitate veteran engagement. According to one participant, BSS “fits well with the culture of the South and how they speak about G-d.” In Bible Belt regions, for example, faith is often a shared worldview among veterans. Many veterans in Southern cultures identify their faith group as their primary social network and support system; personal concerns, especially mental health concerns, are often expressed as spiritual distress in these highly religious subcultures. As a result, such veterans tend to use spiritual language to describe mental health concerns and feel empowered when provided with spiritually integrated care.

#### 3.3.6. Fit with Organizational Process

At sites with histories of group counseling using open, outpatient drop-in attendance as opposed to assigned membership in closed, cohort-based groups, BSS implementation struggled. Veterans presumed that they could drop in to BSS groups as they chose, so they did not attend the BSS group regularly, despite BSS being an eight-session weekly closed group. Providers at sites that frequently use drop-in groups “did not see the need to refer to BSS” because veterans could access other groups at their site without a referral.

Mental health provider distrust of chaplaincy led to lower levels of collaboration with chaplains trained in BSS. It is important to note that distrust between clergy and mental health professionals is not unique to this project, but a broader issue in VA and community care contexts throughout the country [34]. At one unsuccessful site, local mental health referral sources did not understand the program well, resulting in inappropriate referrals, and veterans entering the BSS program with inaccurate expectations.

### 3.4. Implementation

#### 3.4.1. Using BSS as Planned

The settings chosen for BSS were heterogeneous across sites, with some sites placing BSS in locations like Vet Centers, while others chose to locate it in inpatient or outpatient mental health settings. The reasons for the differences varied. For those who chose to locate BSS in Vet Centers, the reason given was resistance from MH staff at the main VA facility. For those who implemented BSS groups in residential treatment settings, the reason given was better rates of participation. BSS leaders in inpatient settings also reported that patients dealing with addiction benefitted a great deal from BSS and thus began regularly extending participation to veterans in treatment for substance use disorders. One psychologist disclosed the decision to use BSS techniques in individual therapy settings, a practice which they said improved the quality of PTSD treatment outcomes, providing veterans a way to receive help for spiritual distress. While that site did not actually start a BSS program, the training had a spillover effect in that it made new skills and resources available for individual therapy.

#### 3.4.2. Training

Many participants had recommendations for enhancing BSS training. One provider we interviewed recommended that “ongoing coaching or consultation would have been helpful.” Other suggestions for improvement at these sites included mentoring between successful and less successful sites (such as a community of practice model). Another participant suggested implementing a “dual track”, or trainings focused on the type of providers using BSS, to better equip chaplains and mental health providers with unique trainings more suited to their respective backgrounds. Additional suggestions included providing aspirational BSS facilitators with an endorsement from the intervention creator to facilitate buy-in from reluctant leadership. Participants suggested a screening process for chaplains who volunteer to lead BSS groups as a way of addressing misconceptions and concerns about religious teaching in a VA setting; this suggestion came from a student chaplain (professional VA staff chaplains are trained to provide pluralistic spiritual care). This finding goes hand-in-hand with findings about how to best train BSS facilitators, specifically with chaplains and mental health providers as co-facilitators. For any discipline, practicing spiritually integrated care in a public health setting, such as the VA, requires the ability to ethically adhere to interventions that are equally meaningful across all faith and non-faith groups. There were also suggestions that coaching or consultation would help facility leaders feel more comfortable that BSS could be successfully adopted, increasing leadership willingness to implement the program.

#### 3.4.3. Incentive

Motivation for beginning BSS across sites was mixed. One site’s participants reported that other evidence-based interventions like PE and CPT were “not working” for MIS, acknowledging the main incentive for BSS as “veterans enjoying the group and finding it effective.” Other sites saw no immediate incentive to implement BSS at their facility, even when it aligned with the training backgrounds of mental health providers at those sites.

#### 3.4.4. Resource Needs

The allocation of proper staff and space resources to execute BSS interventions varied across sites. At less successful sites and sites that abandoned BSS implementation, complaints were broad and ranged from over-burdening staff who were managing high rates of referrals and waiting lists, lack of staff and scheduling issues with MH providers, need for more chaplains to facilitate extra groups, assigning staff to different duties during scheduled group time, lack of standardized assessments for MIS and its absence from the DSM, and lack of provider knowledge about MIS and when to refer veterans. At one site that lacked chaplains to support mental health providers in BSS implementation, a psychologist said they would be “comfortable starting a group if I had a chaplain to do it with me.” Several participants indicated that concerns related to limited staff time were barriers; some providers who wanted to start BSS programs were booked so tightly with clinical work that they did not have the time to carry out the administrative work necessary to start a group, such as establishing clinic codes, finding a place for the group to meet, educating referral sources, and planning a procedure for referrals. Multiple sites felt that they needed more chaplains trained in BSS, either to meet the demand or to work with mental health providers who wanted to collaborate.

On the other hand, at sites with strong chaplaincy and mental health collaboration, shared resources, such as conference/group rooms, contributed to the success of the program. At one successful site, medical support assistants (MSAs) who were assigned to the BSS team helped improve attendance by reminding veterans of BSS group appointments. Facilities that were able to purchase supplemental education resources felt that this contributed to more effective BSS services. In one case, BSS’s appeal to veterans was also a problem; there were far more referrals than trained staff could take, leading to a long wait list. At other sites, the lack of diagnostic criteria/assessments for moral injury made it difficult for staff to make good referral decisions.

#### 3.4.5. Factors Outside of the Organization

Finally, the COVID-19 pandemic had an impact on BSS implementation; it impacted available resources, required that most psychotherapy be provided virtually, and interfered with collecting effectiveness data. Many sites had plans to start BSS programs that were sidelined because the pandemic required staff time, space, funds, and other resources that they had planned to use for BSS, which subsequently had to be re-allocated. Other sites were able to use currently trained staff to provide virtual BSS groups, and some sites started planning ways to use BSS techniques to address moral injury experienced by healthcare providers struggling with pandemic working conditions. One participant indicated that the stress of the pandemic drew many people to look carefully at their values in preparation for end-of-life issues and that BSS was a helpful resource in that context.

## 4. Discussion and Conclusions

Some of the factors that act as facilitators or barriers to BSS facilitation are common across all types of psychotherapy implementation, including sufficiently trained staff with designated time, communication with referral sources, funds to purchase needed patient education materials, and buy-in at administrative and direct care provider levels [35,36]. Professionals disseminating or using the BSS program should be aware that, as a multi-disciplinary intervention, BSS also encounters some unique barriers and advantages. In general, sites that were able to maintain strong collaboration between mental health and chaplaincy disciplines had more access to staff (group facilitators, clerical support, and educated referral sources) and other resources (fund control points for purchasing patient education materials, physical space, and telehealth resources) needed to make the program successful. Many of the recommendations for changes to BSS training were based on the need for collaboration, including specialized training programs for different disciplines. Because, in many cases, mental health providers had little, if any, training in spiritually integrated care, individuals from mental health disciplines expressed a need for more training, discussion, and support from chaplaincy to become comfortable in carrying out this kind of work. It appeared necessary to train fully qualified providers and chaplains; intern/resident level trainees were apprehensive, and trainees were universally unsuccessful in maintaining a program without the help of BSS-trained independent practitioners. Another training recommendation was for longer-term follow-up for BSS facilitators. The types of follow-up requested often included a desire for time flexibility; one participant preferred SharePoint resources (now in place) or partnership with experienced BSS mentors rather than regularly scheduled consultation calls. There was a clear pattern of greater success in implementing BSS in urban sites vs. rural sites. Factors that may be relevant include difficulty recruiting/maintaining full staffing in rural sites; all three rural sites involved in this project described concerns about staffing as relevant to difficulty with implementation. Urban facilities also have more resources to adapt staffing and locations for implementation. For example, some urban sites were able to collaborate with Vet Centers to overcome barriers, adapt BSS to inpatient settings, have clerical support assigned to the program, and leverage training programs to increase provider resources to the program. Overall, there was agreement among successful and unsuccessful implementation sites that the BSS program meets the unmet need for treating moral injury syndrome and that this has an important impact on both veterans and VA staff. Please see Table 2 for a summary of changes that have been made in BSS implementation based on this study.

The implementation barrier and facilitators identified in this project will be used to improve implementation and dissemination effort for the BSS intervention and can also inform similar efforts for other interventions addressing MIS. A useful implementation framework for using these results is the Evidence-Based Quality Improvement (EBQI) framework because it directly fosters an equal stakeholder partnership, which is critical to successful intervention implementation of an evidence-based intervention [37,38]. EBQI methods include researchers working directly with clinical and administrative decision makers to adapt intervention procedures for specific settings using top-down and bottom-up engagement strategies. Researchers contribute knowledge of the evidence base, training, and fidelity and facilitate problem solving. Clinicians and administrators contribute knowledge of the local context, barriers, and facilitators and access to resources to facilitate implementation. An important component of EBQI for implementing evidence-based interventions is the use of Plan–Do–Study–Act cycles whereby collaborating researchers and clinicians can adjust implementation strategies to maximize implementation outcomes [39].

More specifically, for individuals involved in implementation of BSS, results from this project will inform the following EBQI components. For example, engaging leaders and frontline clinicians provides an opportunity to identify site-specific barriers that can be addressed iteratively throughout the implementation process and at the same time build site-specific intervention buy-in. Evidence of intervention effectiveness is needed both from the literature and from local site evaluations to provide data to support implementation efforts. Adapting the intervention is needed to better fit the intervention with local conditions and resources. Adaptation examples included the location or setting for BSS groups (e.g., medical center or Vet Center); facilitation of groups by chaplains, mental health providers or both; and adding mindfulness components or maintenance sessions. Enhancing confidence and skills included making sure that staff were trained in addition to trainees and adding BSS coaching or booster training sessions.

As with any project, this project has strengths and weaknesses. Strengths of this project include use of CFIR to guide data collection, analysis, and interpretation; geographic variation in sites; qualitative data collection and analysis to provide a more detailed understanding of barriers and facilitators; and representation of participants from diverse races, faith identifications, disciplines, and levels of training. Limitations include the small number of interviewees, and that data collected exclusively in the VA system may not be generalized to non-VA facilities. A BSS facilitator trainer being known to all of the participants may also have introduced social desirability into responses. A more comprehensive study would have also collected data from veterans, as well as service chiefs and facility leadership.

While there is much more work to be carried out to inform implementation of MIS treatment within the VA system, BSS represents one asset that can contribute to resolving this problem. No intervention will work for every veteran, and research currently being conducted on Adaptive Disclosure, the Impact of Killing in War, Moral Elevation, the Self-Forgiveness Workbook, and Trauma-Informed Guilt Reduction is essential and should continue [19,20,21,25,26]. Given the current rate of scaling of BSS, it would be important to begin larger scale implementation studies, not only about BSS specifically but about systemic approaches to assessing and treating MIS. Such studies may make important contributions to high-priority VA goals, such as suicide prevention and personalized, whole health care.

## Figures and Tables

**Table 1 behavsci-14-00281-t001:** Primary project findings.

MODEL COMPONENT	FACILITATORS	BARRIERS
**INTERVENTION** **CHARACTERISTICS**		
Why BSS?	One site chose BSS because other group protocols for chaplains did not work for vets with PTSD. “BSS was more welcoming [than other spiritually integrated approaches], with more universal language.”	Moral injury not in the DSM, so MH providers do not know when to DX/refer for MI treatment.
Evidence needed	One chaplain could have carried out the intervention independently but shared slides about the evidence base with mental health staff. “We could not implement BSS in the main facility because MH said it was not ‘evidence based,’ so we worked with the Vet Centers instead.”	**In multiple sites, a single psychologist interfered with implementation because it was not “evidence-based” [note—BSS is in fact evidence-based]** Chaplains had difficulty articulating the evidence base for BSS when they met resistance from MH. One site stated that they needed more local program evaluation data. “It was difficult to get data showing effectiveness (i.e., doing measurement instruments) with virtual groups in the pandemic.”
Changes needed	One site added mindfulness exercises to the beginning of each session. One site added an additional session to build trust before proceeding through the protocol. One site added more sessions on forgiveness and follow-up sessions at the end.	**Multiple sites would recommend BSS follow-up sessions to help maintain therapeutic effects over a longer term.** **Some sites do not emphasize the Good Goats book**.
Local Effectiveness	One participant reported the strong need to use this for suicide prevention for PTSD, addiction, and inpatient psych. “One of my veterans thought of himself as a killer …. The group changed his identity to an angel of mercy.” BSS helped veterans gain self-compassion early in the group.	Mental health providers tasked with referring veterans did not understand the program, resulting in bad referrals and inaccurate expectations. “It was too difficult to gather outcome measures in virtual groups. We could have gotten more support for the program if we had that data.”
**INDIVIDUAL CHARACTERISTICS**		
Disciplines Trained	In most sites, chaplains were originally trained and subsequently lead the group. A large number of mental health providers trained with a previously trained chaplain. “All chaplains should be trained in BSS.”[Chief Chaplain]	One site described a neo-orthodox chaplain student managing theologies that can be used judgmentally as a barrier to effective implementation. At one site, a student chaplain was unable/unwilling to adhere to manual and did not keep to the ecumenical program. Students were apprehensive about leading BSS groups without a supervisor present.
Group Leaders	**Chaplains often had a mental health co-leader.**	**Multiple unsuccessful sites reported insufficient staffing for group leadership as a barrier to implementation.**
Confidence	We found the protocol very effective. **Vets wanted to take part in more sessions.** One participant reported confidence in their use of this intervention only after a few groups. “I found it fairly easy to work through the program [psychologist].” “I was very confident about using BSS [chaplain specializing in mental health].”	“I did not have confidence that I could do the group at first, but that was just me; the training was fine.” One participant reported that they needed more help than provided in the training [this was a chaplain with no experience working with manualized intervention]. They would like a sharepoint [resource available now]. “I was not confident; I had not worked with spiritually integrated care before. I was self-conscious about spiritually integrated care because I am not Christian, not because the training was insufficient [social worker].”
**ORGANIZATIONAL FACTORS**		
Team involved	One successful team included a psychologist/suicide prevention specialist, a psychologist/local recovery coordinator, and a chaplain clinical pastoral education supervisor who supervised chaplain residents. Mental health providers were successful in introducing the evidence base and advocating for use of BSS; chaplains needed collaboration with MH providers to start the program. A chaplain who was experienced in cofacilitation with MH providers was very successful in bringing BSS to multiple cohorts of veterans.	**Sites where only students (interns, CPE students) were trained were universally unsuccessful in starting programs**.
Organizational receptivity	BSS is considered a “best practice” at Vet Centers. Sites that met resistance at a VA facility were able to implement at Vet Centers. BSS training “explains the significance of ‘spirituality’, parsing it out to chaplains and therapists that this touches a vets personhood in an important way.” One participant recommended that trainers should ”remind non-religious MH providers that they do not have to speak from a G-d perspective; the mechanism is integrating many perspectives, so any perspective will help.”	There was more resistance from MH providers earlier in the history of treatment for MI (2011–2016). MH providers are spiritual at very low rates and tend to be resistant to implementing spiritually integrated care. **MH providers were uncomfortable with spirituality.** **Facility does not value chaplaincy.** Facility had no intention of starting a program; simply joined training because the local Vet Center was receiving the training. MH provider/planned group coleader was so uncomfortable with spirituality that they wanted to remove all spiritual language from the protocol.
Leadership receptivity	BSS training helped individuals make the case for using BSS in leadership. **Chaplain chiefs who are familiar with Dr. Harris’s work were very interested in supporting BSS programs**.	A leader who was receptive to BSS implementation left the facility. Because the “VA is very militaristic… [there] should have been some ‘commander to commander’ talk so the program would have come in with more backing.” “For other Evidence Based Psychotherapies, you have to apply to do the training, and administrators sign off on a commitment to give you time to do the training, consultations, and the group. That would help build accountability for the organization. A consultation phase afterwards would also help build accountability for the organization.”
Organizational openness to innovation	Many MH staff were grateful to have help with spiritual concerns and very willing to refer	One veteran complained; subsequently the program was shut down without any communication to BSS group leaders as to why. **Chaplains and/or mental health professionals who had developed other approaches to addressing moral injury discouraged use of BSS. This was the case even if there were no randomized controlled trials to support use of the other intervention.** “Psychologists were threatened by MI treatments;” they feared that chaplains were “trying to horn in on what they were doing.” Student chaplain who started group “felt that I was being driven out.” **Some chaplaincy residents did not like the Good Goats book**.
Perceived need for BSS	BSS is comparable to CPT and CBT and is a superior intervention for people who do not want mainstream PTSD treatments. **BSS is a very needed service.** There is a growing need for MI treatment even among providers in post-COVID settings.	No sites denied the need for BSS services.
Fit with organizational process	The group format is needed for healing; one-on-one would not work. **Better attendance in inpatient settings.** “Fits well with culture of the South and how they speak about G-d in the region, allowed [patient] to talk through ‘being angry with G-d’ in way they understood”.	**The organization has history of little or no group counseling services.** Organization has history of only open groups, so vets did not see the need to attend regularly, and providers did not see the need to refer. In Bible Belt culture, some faiths distrust science; Some MH providers and chaplains distrust one another in that culture. Culture in the Northeast is less religious, making it harder to find providers to implement BSS. “I shared a presentation with mental health, but got no referrals because they thought it was an open group.” “We are [a rural VA]. Many of my veterans tell me they moved here so it would be easier to isolate themselves. It is difficult to recruit these veterans into groups.”
**IMPLEMENTATION PROCESS**		
Using as planned	One site reported their best participation in inpatient psych. Inpatients benefitted (inpatient treatment for addiction rather than serious mental illness). One trained facilitator planned to use BSS in one facility but transferred to another. Individual BSS facilitators used concepts from BSS techniques in individual therapy. This included increased spiritual assessment and inviting veterans to share faith perspectives in PTSD/moral injury treatment.	One site implemented at a Vet Center rather than a VA hospital because MH providers at Vet Centers did not resist, while sometimes MH providers at VA medical centers did. **Multiple sites used BSS with addictions as well as PTSD**.
Training	**Individuals trained in one facility subsequently invited those from other facilities to undergo training**. **Multiple trainees asked for more training for work with atheist/agnostic clients.** One trainee requested a mentorship connection with another site stating that they would prefer this to a monthly meeting. “Dr. Harris could provide a letter for chaplains to share with providers to show they have been trained and are knowledgeable and “give them her stamp.” One participant recommended “dual track training:” one track for MH providers and one for chaplains.	At one site, mental health providers were trained, but no chaplains at that site were trained. MH providers were not comfortable performing the intervention without chaplaincy help. **What could Dr. Harris and her team have done to make it easier to implement BSS at your facility?** “Meeting with psych staff and leadership to tell them what the point of BSS is and the role of chaplains in it.” “Might need to carefully screen the chaplains who are involved to make sure they don’t do things the psych’s are worried about” [Note that this concern came from a student chaplain.] “In certain states one would have to give off the record assurances that [chaplains] wouldn’t come in and sabotage [MH providers’] work.”
Incentive	**Evidence-based interventions like PE and CBT are not working on moral injury.** The main incentive is the vets enjoying the group and finding it effective.	Other than its consistency with my theoretical orientation, there is no incentive to take part in BSS. There is no immediate incentive to start a BSS group. **Aside from increasing patient care, there are no incentives**.
Resource needs	Used MH conference rooms. Another benefit at [site] was the high level of MSA support; MSA’s support helped improve attendance. Veterans found the Good Goats book very healing.	**Not enough staff.** I would like a second chaplain trained to meet the demand for BSS. **There are no standardized assessments for moral injury.** Outpatient psychologists scheduled too tightly to do the administrative work to start the group. “I would be comfortable starting a group if I had a chaplain to do it with me.” **Ongoing coaching or consultation would have been helpful.** Too many referrals-forcing BSS leaders to maintain a wait list for groups. We had no group rooms.
Factors outside of organization	Pandemic requires some sites to make groups virtual, but still perceived as effective. Growing need for MI treatment even among providers in post-COVID settings. “Pandemic put a light on loneliness, meaning, estrangement, end of life, necessity of having things in place in all lanes.”	**The COVID-19 pandemic interfered with plans to implement BSS.** “It was too difficult to gather outcome measures in virtual groups. We could have gotten more support for the program if we had that data.”

Bolded items reported at multiple sites.

**Table 2 behavsci-14-00281-t002:** Changes made to BSS based on these data.

Concern Addressed	Change Made
BSS not perceived as evidence-based	Increased attention to research supporting BSS in training; Had MH professionals on the BSS dissemination team collaborate with chaplains in communicating about the evidence base for BSS to MH; Published additional empirical evidence for the theoretical basis of BSS; Obtained funding for additional clinical trials; Assisted sites with identifying and reporting on measurement-based care outcome instruments.
Desire for additional, ongoing training/ consultation for new BSS leaders	Paired willing sites with mentor sites; Offered as-needed consultation with the BSS dissemination team; Applied for funding to pay for additional ongoing consultation.
Fears about working with individuals from diverse spiritual groups, especially agnostic/atheist	Updated training to include specific techniques and inclusive language; Increased training on ethical practices for spiritually integrated care in public settings.
Application of BSS in inpatient settings	Amended training to address ethical practice in inpatient settings
Desire for more contact/additional sessions/additional support especially for forgiveness	Made the “Self-Forgiveness Workbook” available on the BSS website as an optional addition to BSS (Griffin et al., 2015) [25]

## Data Availability

Data presented in this study are available on request from the corresponding author. Any shared data will be de-identified to standards specified by VA privacy policies.

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
