# Peer review of "Implementing a Multi-Disciplinary, Evidence-Based Resilience Intervention for Moral Injury Syndrome: Systemic Barriers and Facilitators"

_behavsci, 2024, doi:10.3390/bs14040281_

Round 1

Reviewer 1 Report

Comments and Suggestions for Authors

The article is very interesting. Several questions arise.

1.In line 39, the authors write that "They may also witness or be the victim of others violating these standards (Stein et al. 2012). Shouldn't it be written in the past tense or add to the literature?

 2.In lines 46 to 68, the authors write about percentages or post-9/11 veterans. Shouldn't one write only about percentages?

3.The methodology section clearly lacks the exact date of the study, from when to when.

4.In line 452 the authors write that the pandemic had an impact on BSS implementation. The sentence is very general, it is well known that the pandemic changed all social, economic, religious, personal life. Therefore, for what purpose such a general sentence.

5.The title of the fourth part is Conclusions and Discussion. Shouldn't it be the other way around, discussion first and then conclusions. 

6 The reviewer doesn't really understand with whom the authors are discussing in lines 466 to 492.

7 The situation is similar in lines 511 to 522.

Comments on the Quality of English Language

The text is understandable

Author Response

Thank you to you and to your reviewers for the opportunity to improve our submission, “Implementing a Multi-Disciplinary, Evidence-Based Resilience Intervention for Moral Injury Syndrome: Systemic Barriers and Facilitators.”  Here is a summary of our response to the reviews.

You noted that 11 of our citations came from within the research team.  We looked closely at this.  The field of moral injury treatment is nascent, and the literature is growing, but still small.  As a result, many of the citations of work from team members could not be replaced with other work.  We succeeded in finding three citations from within the team for which we could find acceptable citations from outside the team, and we have removed those three citations.

You appropriately noted that we had omitted required back matter.  We have added that material to this version of the manuscript.

You appropriately noted that our references were not formatted consistently with the journal’s instructions.  We have corrected this.

Reviewer 1 noted that the tense in line 39 should be in past tense.  This has been corrected.

Reviewer 1 asked us to clarify findings; some of the research cited gave percentages of veterans managing PTSD who also have moral injury syndrome.  Other research only sampled post 9/11 veterans , and gave odds related to suicide risk in that cohort.   We agree that the juxtaposition of percentages, odds, and different cohorts can be confusing.  Because different cohorts of veterans present with different medical needs, we did not feel that we could reduce this to a single percentage that would accurately represent prevalence because there were different samples involved.  However, to reduce confusion, we converted odds into percents so that readers would have fewer mathematical conversions to consider when looking at the prevalence data available.

Reviewer 1 asked for the dates in which data was collected.  We have added those dates.

Reviewer 1 asked that a statement about the impact of the pandemic on BSS implementation be developed to be more specific; we have added clauses to that sentence to make the sentence more meaningful.

Reviewer 1 noted two parts of the discussion in which they were unsure “with whom the authors are discussing.”  We have added a sentence to both of these paragraphs to specify the audience the paragraph was to target.

Reviewer 2 noted line and paragraph spacing irregularities.  We have corrected these as much as possible. Note that the journal has changed the article to be both right and left justified, and often this results in spacing irregularities.

Reviewer 2 noted that some question marks had been retained in headings and in tables.  We have removed these.

Reviewer 2 noted a reference formatting concern.  This has been corrected, and references throughout have been reformatted to be consistent with the journal’s instructions to authors.

Reviewer 2 appropriately noted that the interviewer was known to participants and may have brought demand characteristics into play.  As this project was unfunded, we did not have the option of hiring alternative staff to gather this quality assurance data. We have noted this as a limitation in the discussion.

Reviewer 2 requested definitions of “successful” and “unsuccessful” sites.  We have added such a definition.

Reviewer 2 makes a very appropriate request for a table outline the demographics for participants; this is routinely done for such studies, and in an earlier draft of the article, we had such a table.  Because of the small sample, and the necessary description of sites, when we had this table, demographic data could readily be attached to sites, making many, if not all, of the participants identifiable to readers from within the VA System.  Our author team discussed this and determined that ethically we are limited to summarizing the demographics and numbers in narrative form to protect the confidentiality of participants.  Note that the first sentence of the “Participants” section notes a total of seven participants, so the total N is specified.

Reviewer 2 made several excellent suggestions for improving the effectiveness of Table 1.  This has been significantly reformatted.

Thank you for the opportunity to revise and improve this article.  We look forward to hearing of your editorial decision.

Reviewer 2 Report

Comments and Suggestions for Authors

This is a clear, well written article that highlights the important topic of MIS that is not only important in veterans with PTSD but also other survivors of war. There also appears to be a slightly heavier emphasis on the challenges on implementation. 

·        There appears to be some spacing issues between the periods and new sentences throughout the article (For example: Line 113, 115, 138, 503) there also appears to be some formatting issues between paragraphs (Line 492-495).

·        Line 126-127 – it appears that interviews were completed by a BSS trainer who completed prior training at that site. Were there any measures taken to make sure the interviewees didn’t want to “impress” or “embellish” the interviewer since there was an already known relationship?

·        Line 136-137 – it may strength the article by listing the definition/criteria that authors used for “success” and “unsuccessful” implementations of the BSS programs.

·        Line 159 – I believe it would strengthen the article to have a clear “N” and demographic description table. Currently as written it is not as clear to readers about participants. 

·        Line 384 appears to have a question mark in the heading

·        Line 509 appears to have a formatting reference issue

·        Line 760 – this is a very much appreciated table that summarizes results. It appears to have odd spacings between tables, question marks left in, and has both completed sentences and phrases. It can be strength by having consistency in structure.

Comments on the Quality of English Language

Appropriate English language, however consistency is needed in the table at the end of the article (full sentences vs. paraphrases/incomplete sentences).

Author Response

(The authors gave the same response as above.)

Round 2

Reviewer 2 Report

Comments and Suggestions for Authors

Authors have fully addressed my comments from prior submission!